# Unleashing Multispectral Video's Potential in Semantic Segmentation: A Semi-supervised Viewpoint and New UAV-View Benchmark

**Wei Ji**[*1,2], **Jingjing Li**[*1], **Wenbo Li**[†3], **Yilin Shen**[3], **Li Cheng**[1], **Hongxia Jin**[3]

*Project website:* https://jiwei0921.github.io/MVUAV

## Abstract

Thanks to the rapid progress in RGB & thermal imaging, also known as multispectral imaging, the task of multispectral video semantic segmentation, or MVSS in short, has recently drawn significant attentions. Noticeably, it offers new opportunities in improving segmentation performance under unfavorable visual conditions such as poor light or overexposure. Unfortunately, there are currently very few datasets available, including for example MVSeg dataset that focuses purely toward eye-level view; and it features the sparse annotation nature due to the intensive demands of labeling process. To address these key challenges of the MVSS task, this paper presents two major contributions: the introduction of MVUAV, a new MVSS benchmark dataset, and the development of a dedicated semi-supervised MVSS baseline - SemiMV. Our MVUAV dataset is captured via Unmanned Aerial Vehicles (UAV), which offers a unique oblique bird's-eye view complementary to the existing MVSS datasets; it also encompasses a broad range of day/night lighting conditions and over 30 semantic categories. In the meantime, to better leverage the sparse annotations and extra unlabeled RGB-Thermal videos, a semi-supervised learning baseline, SemiMV, is proposed to enforce consistency regularization through a dedicated Cross-collaborative Consistency Learning (C3L) module and a denoised temporal aggregation strategy. Comprehensive empirical evaluations on both MVSeg and MVUAV benchmark datasets have showcased the efficacy of our SemiMV baseline.

## 1 Introduction

Semantic segmentation is the process of categorizing each pixel in an image/video to a specific class label, which plays a vital role in visual scene understanding [1, 2]. Remarkable progresses has been made in the past several years, particularly in RGB-based semantic segmentation [3, 4, 5, 6, 7, 8, 9, 10, 11]. With the increasing accessibility of thermal sensors in capturing thermal radiation of objects with temperature above absolute zero, multispectral semantic segmentation (MSS) [12, 13, 14, 15], where both RGB and infrared thermal cameras are engaged, starts to gain notable traction. An exemplar illustration is shown in Fig. 1. This multispectral imaging setup excels specifically in scenes with unfavorable visual conditions, such as low-light or overexposure. On the other hand, the dynamic and ever-changing nature of real-world scenarios has propelled interests in video semantic segmentation (VSS) [16, 17, 18, 19, 20] with impressive segmentation performance. The task of multispectral (aka RGB-Thermal or RGB-T) video semantic segmentation (MVSS)[21] has emerged recently as a promising visual segmentation setup that has the best of both worlds.

Nevertheless, as being a relatively new task, there are still a number of hurdles in MVSS research. The most prominent one is the lack of quality datasets. One major MVSS benchmark presently

---

[1]University of Alberta [2]Yale University [3]Samsung AI Center-Mountain View (work done here). [†]Intern mentor. [*]Equal contribution. Corresponding author: Jingjing Li <jingjin1@ualberta.ca>.

38th Conference on Neural Information Processing Systems (NeurIPS 2024).

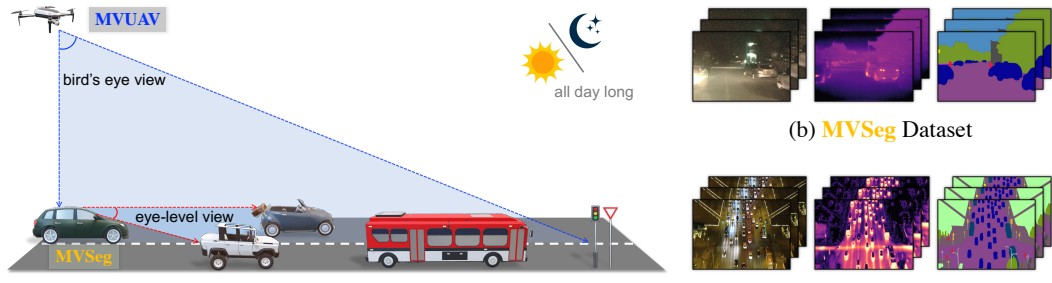

(a) Data Acquisition

(b) **MVSeg** Dataset

(c) **MVUAV** Dataset

Figure 1: (a) Viewpoint diversity of the existing MVSeg dataset [21] and the new MVUAV dataset. (b) & (c) Representative samples from the MVSeg & MVUAV datasets, where RGB videos, thermal videos, and the corresponding semantic annotations are visualized.

available is MVSeg [21], a recently introduced dataset. It is worth mentioning that the multispectral videos in existing MVSS datasets, including MVSeg, are all taken from eye-level views, which clearly suggests the lack of data source diversity in MVSS. Even after securing good quality of RGB-T video footage, it is still a demanding exercise to make annotations. In practice, this is achieved by furnishing pixel-wise semantic labels on a selected subset of frames or key-frames[1]. Consequentially, there typically exists only *sparse* annotations in the MVSS datasets.

This paper aims to address the above-mentioned challenges. Specifically, we introduce MVUAV, a new MVSS dataset containing a diverse range of RGB-T videos captured by Unmanned Aerial Vehicles (UAVs) from an oblique bird's-eye viewpoint. As depicted in Fig. 1, this viewpoint provides a broader, holistic perspective free from the constraints of eye-level capture adopted by existing MVSS datasets such as MVSeg. In detail, our MVUAV dataset comprises 413 RGB-Thermal videos with 53,828 frame pairs in total. A subset of 2,183 key-frame pairs are meticulously annotated with pixel-wise semantic labels across 36 different classes of objects and stuffs, where the pixel annotation rate is 99.18%. Our dataset captures diverse real-world scenarios such as roads, streets, bridges, parks, seas, beaches, courts and schools; it also spans different lighting conditions from daytime to low-light and even pitch-dark scenarios. The MVUAV also presents some challenges, such as large scale variation, fine-grained scene parsing, moving object segmentation, and adverse illumination conditions, making it a valuable asset for evaluating various MVSS algorithms.

Meanwhile, we seek to tackle the MVSS task from a relatively new semi-supervised perspective. As illustrated in Fig. 2, our strategy utilizes a small number of sparsely labeled RGB-Thermal videos, alongside massive amount of unlabeled videos. In the closely-related domain of semi-supervised RGB-based semantic image segmentation, the idea of consistency regularization has played important role in the eventual performance. Its efficacy stems from perturbation-invariant training, by enforcing consistent predictions despite of varied perturbations presented in unlabeled RGB images at different processing levels: input [22], feature [23, 24], or network [25]. There motivates us to explore consistency regularization in RGB-Thermal videos. In fact, MVSS seems to be an ideal setting, since RGB and thermal images essentially capture the same scenes from different sensory perspectives, that offers innate input perturbations. Further, processing the multimodal data through two parallel networks with distinct parameters also leads to valuable feature-level and network-level perturbations. Building on these insights, we introduce SemiMV, a novel semi-supervised MVSS framework. At its core is a Cross-collaborative Consistency Learning (C3L) module, where the RGB and thermal streams mutually offer pseudo-supervisions to each other. A pixel-wise reliability map is also generated, based on the learned cross-modal consistency, to guide the temporal fusion process and mitigate potential noise.

The main contributions of this paper are summarized as follows:

- A new benchmark dataset, MVUAV, is introduced. The RGB-T videos in the new dataset present complementary perspectives to the the existing MVSS datasets such as MVSeg that are typically from eye-level views, by capturing from an oblique bird's-eye viewpoint. It also provides pixel-level dense annotations with a rich set of 36 visual semantic categories.

---

[1]In video-based segmentation datasets (*e.g.,* Cityscapes [1] and MVSeg [21]), it is a common practice to *sparsely* annotate video frames, given similar content of consecutive frames and substantial cost saving in the amount of annotation effort.

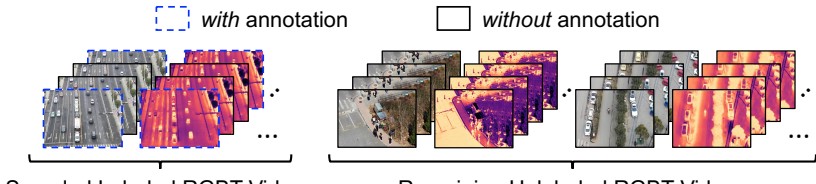

Figure 2: Illustration of training examples in semi-supervised MVSS setting, where a limited amount of sparsely labeled RGB-Thermal (RGBT) videos and massive unlabeled ones are utilized.

- We propose a simple yet effective semi-supervised MVSS baseline, SemiMV. Our baseline is to our knowledge the first in employing the consistency regularization idea tailored for the semi-supervised MVSS task. Our experimental evaluation on the MVSeg and MVUAV benchmark datasets demonstrates the efficacy of the SemiMV baseline.

## 2 Related Work

### 2.1 Semantic Segmentation in Diverse Modalities

Semantic segmentation, a crucial task in computer vision, has evolved significantly over recent decades. This evolution spans four primary data modalities: RGB images, multispectral images, RGB videos, and multispectral videos, each uniquely contributing to the field's advancement and expanding its practical applications [26, 27, 28, 29, 19, 30, 31, 32].

*RGB semantic segmentation (RSS)* has undergone remarkable advancements. FCN [33], as a milestone work, uses a fully convolutional network for per-pixel prediction. Subsequent innovations, such as dilation convolution [34], pyramid pooling [35], and attention mechanisms [36, 37, 38], have enriched segmentation by integrating contextual information and capturing global contexts. Recently, transformer-based methods [39, 40, 41, 42] have gained prominence with the advent of Visual Transformer [43]. For a detailed overview, readers are advised to refer to comprehensive survey papers [44]. *Multispectral semantic segmentation (MSS)* is an emergent field that aims to enhance model robustness in adverse lighting conditions by integrating RGB and thermal imagery [45]. The primary challenge in MSS lies in effectively merging RGB and thermal data. Various strategies have been developed, such as concatenation [46, 47], element-wise summation [48, 49], bridging-then-fusing [14], attention-weighted fusion [50], explicit extract and fusion [13], and cross-modal fusion [15]. Additionally, there is a related area of research focused on multimodal RGBD-based segmentation [51, 52, 53, 54, 55, 56, 57, 58, 59, 60, 61, 62, 63, 64, 65, 66], which addresses some limitations of purely RGB-based segmentation. On the other hand, *video semantic segmentation (VSS)* is gaining traction with the growing importance of dynamic video processing. VSS models [18, 67, 16, 68, 69, 70] have focused on leveraging temporal contexts of video frames. Notably, some studies [71, 72, 73] utilize optical flow [74] to warp features from neighbouring frames for feature alignment and aggregation. Recently, attention-based methods [17, 75, 76, 77] have been proposed to selectively retrieve information from past frames for improving current frame segmentation, yielding promising results. *Multispectral video semantic segmentation (MVSS)* is a nascent area that combines the strengths of multispectral and temporal contexts. A pioneering study [21] introduces the large-scale MVSeg dataset to benchmark this field, and proposes a baseline model for learning a joint representation from multispectral video input. Our paper contributes a new MVUAV dataset to enrich the dataset diversity in MVSS.

### 2.2 Semi-supervised Learning in Segmentation

Semantic segmentation has a major challenge in real-world scenarios where only limited pixel-level labels are available due to high expense of human labor. Recently, semi-supervised semantic segmentation has surged in popularity to utilize a plethora of unlabeled data.

*Semi-supervised RSS* has gained extensive research interest by extending the powerful semi-supervised learning (SSL) techniques. Notably, consistency regularization and self-training have demonstrated great success. Consistency regularization enforces the consistency of the predictions with various perturbations, *e.g.*, input perturbation via augmenting input images [22], feature perturbation

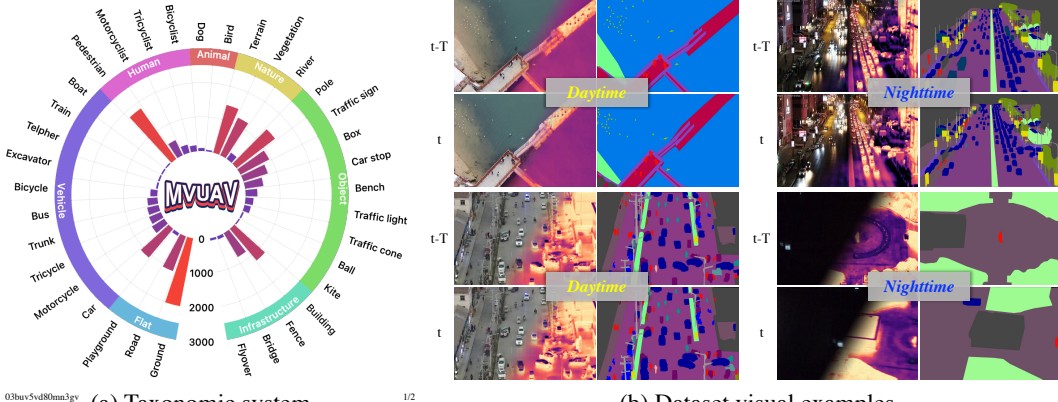

(a) Taxonomic system                    (b) Dataset visual examples

Figure 3: Illustrations of the proposed MVUAV dataset. (a) Taxonomic system and its histogram distribution showing the number of annotated frames across different categories. (b) Examples of multispectral UAV videos and corresponding annotations in both daytime and nighttime scenarios.

using multiple decoders [23] or feature dropout [24], and network perturbation through different initialization [25]. Meanwhile, self-training methods [78, 79] generate pseudo segments for the unlabeled images and train subsequent models with both human-annotated and pseudo-labeled data. To leverage temporal video information, several semi-supervised VSS methods [78, 80] have been devised. For example, Zhuang *et al.* [80] propose inter-frame feature reconstruction to leverage the ground-truth labels to supervise the model training on unlabeled frames. Despite their effectiveness, these semi-supervised models are limited to single-mode RGB inputs. *To our best knowledge, our work is the first to deal with semi-supervised MVSS problem with RGB-Thermal video inputs.*

## 3 The MVUAV Dataset

This section describes the construction of the MVUAV dataset and analyzes its statistical results.

**Dataset Collection.** The main principle of data acquisition is to provide a comprehensive collection of calibrated RGB and thermal infrared video sequences from a new UAV perspective, and furnish high-quality semantic annotations. Toward this objective, we initially gather about 500 RGB-T video sequences from a latest aerial tracking dataset VTUAV [81] (altitudes ranging from 5-20m). The RGB-Thermal videos are captured at diverse environments such as parks, streets, and beaches, under various lighting conditions and seasons. To ensure the quality of our dataset, we remove unqualified videos or frames that are blurry, misaligned, or repetitive. After this selection process, the MVUAV dataset consists of 413 high-quality multispectral UAV videos, with 53,828 paired frames in total.

**Dataset Annotation.** We then provide pixel-wise semantic labels to the multispectral UAV videos. We employ Labelme toolkit to annotate the multispectral UAV videos. The annotation process presents many challenges. First, the dataset has many complex scenes at adverse conditions, *e.g.*, nighttime, darkness, overexposure, making it difficult to identify target objects from RGB images alone. Second, it is crucial to maintain annotation consistency across video frames, otherwise objects might be misidentified in different frames. Third, the UAV-view presents a more complex and crowded visual field compared to the eye-level view in MVSeg, making object identification and silhouette distinction more challenging. To overcome these challenges, we first overlay thermal heatmaps onto corresponding RGB images to visually aid annotators in identifying objects in complex scenes. Meanwhile, video frames from the same video are annotated by the same person to ensure temporal consistency. Three inspectors review and correct the annotations to make sure temporal consistency, with the assistance of video-level annotation visualization. Due to these challenges, the annotation and quality control process takes about 90 minutes per frame on average. We visualize some representative examples from the MVUAV dataset in Fig. 3.

**Dataset Statistics.** Table 1 outlines some critical attributes of the MVUAV dataset and related semantic segmentation datasets with different modalities. Our MVUAV dataset comprises 413 RGB-Thermal videos at a frame rate of 25 fps, including 54k image pairs in total and 2,183 annotated image pairs. The new MVUAV dataset can act as a valuable asset to complement the existing MVSeg dataset for more thorough evaluations of various MVSS models. Additionally, the MVUAV dataset

Table 1: Statistics of various semantic segmentation datasets in diverse modalities. 'Surv.', '#Cls' and 'Anno.' are the shorthand for surveillance, the number of classes and annotation density, respectively.

| Dataset | Year | Color | Infrared | Video | UAV | Capture | #Vids(Frames) | #GTs | Resolution | #Cls | %Anno. |
|---|---|---|---|---|---|---|---|---|---|---|---|
| Cityscapes [1] | 2016 | ✓ | ✗ | ✓ | ✗ | Car | - (150k) | 5,000 | 2048×1024 | 30 | 97.10% |
| CamVid [82] | 2009 | ✓ | ✗ | ✓ | ✗ | Car | 5 (40k) | 701 | 960×720 | 32 | 96.20% |
| UAVid [83] | 2020 | ✓ | ✗ | ✓ | ✓ | Drone | 42 (38k) | 420 | 3840×2160 | 8 | 82.69% |
| SODA [84] | 2020 | ✗ | ✓ | ✗ | ✗ | Pedestrian | - | 2,168 | 640×480 | 21 | 79.73% |
| SCUT-Seg [85] | 2021 | ✗ | ✓ | ✗ | ✗ | Car | - | 2,010 | 720×576 | 10 | 56.50% |
| MFNet [46] | 2017 | ✓ | ✓ | ✗ | ✗ | Car | - | 1,569 | 640×480 | 9 | 7.86% |
| PST900 [47] | 2020 | ✓ | ✓ | ✗ | ✗ | Robot | - | 894 | 1280×720 | 5 | 3.02% |
| SemanticRT [45] | 2023 | ✓ | ✓ | ✗ | ✗ | Surv. | - | 11,371 | 1280×1024 | 13 | 21.27% |
| FMB [86] | 2023 | ✓ | ✓ | ✗ | ✗ | Car | - | 1,500 | 800×600 | 15 | 98.16% |
| CART [87] | 2024 | ✓ | ✓ | ✗ | ✓ | Drone | - | 2,282 | 960×600 | 11 | 99.98% |
| MVSeg [21] | 2023 | ✓ | ✓ | ✓ | ✗ | Car/Surv. | 738 (53k) | 3,545 | 480×640 | 26 | 98.96% |
| **MVUAV** | - | ✓ | ✓ | ✓ | ✓ | Drone | 413 (54k) | 2,183 | 1920×1080 | 36 | 99.18% |

features high-resolution imagery (1920×1080), which is helpful for fine-grained scene parsing. This dataset also provides detailed annotations for a rich set of semantic categories, covering 8 root categories and 36 sub-classes (including background) as shown in Fig. 3, at a high pixel annotation rate of 99.18%. This can facilitate detailed and comprehensive scene understanding. *Additional dataset details and discussions with related UAV-view datasets [88, 89, 90, 91] are provided in the supplementary material.*

**Dataset Splits.** The dataset is partitioned into training, validation, and test sets, which contain 275, 35, and 103 videos respectively, with 1,464, 171, and 548 annotated masks. One frame is annotated for every 25 frames. Additionally, the test set data involves both daytime and nighttime scenes, consisting of 72 videos with 351 annotated frames and 31 videos with 197 annotated frames, respectively.

## 4 Methodology

### 4.1 Problem Definition

The training setting of the semi-supervised MVSS task is illustrated in Fig. 2, where we have a small set of labeled multispectral videos with sparse annotations and a larger corpus of unlabeled multispectral videos. Following the common practice in VSS [75, 17] and MVSS [21], we use video clips as input units. Each video clip contains a sequence of $t$ frame pairs and only the final frame pair in the labeled video clip has semantic annotations. Formally, we denote a multispectral video clip as $\mathcal{V} = \{(I_i^R, I_i^T)\}_{i=1}^t$, where $(I_i^R, I_i^T)$ represents the $i$-th frame pair with spatial resolution of $H \times W$. The labeled set is denoted as $\mathcal{D}^L = \{(\mathcal{V}_n^L, y_n)\}_{n=1}^{n_L}$, which comprises $n_L$ clips, and $y_n$ is the pixel-level semantic labels for the final frame pair of each clip, in a space of $C$ classes. The unlabeled set is denoted as $\mathcal{D}^U = \{\mathcal{V}_n^U\}_{n=1}^{n_U}$, including $n_U$ unlabeled multispectral video clips. Additionally, we use an evaluation set, $\mathcal{D}^V = \{(\mathcal{V}_n^V, y_n)\}_{n=1}^{n_V}$. The goal of Semi-MVSS is to develop a segmentation model that can effectively learn from both $\mathcal{D}^L$ and $\mathcal{D}^U$, and exhibit robust generalization to $\mathcal{D}^V$.

### 4.2 Proposed SemiMV Framework

Fig. 4 depicts the overall architecture of our SemiMV. The network takes a multispectral video clip as input, which contains a Query frame pair at time step $t$, and $M$ Memory pairs selected from past frames. In the semi-supervised MVSS setting, only the Query pairs from the labeled video set have ground-truth semantic annotations.

**Supervised Training.** We first feed the RGB and Thermal pairs into two parallel segmentation networks ($Net^R$ and $Net^T$), *e.g.*, DeepLabv3+, which generate initial segmentation predictions $P_i^R$ and $P_i^T$ ($i \in [t-M, \cdots, t]$). *For the labeled Query images*, common supervised training is employed on the outputs of two networks using ground-truth segmentation maps, represented by:

$$\mathcal{L}_{sup} = \mathbb{E}_{(I_t^R, I_t^T, y) \in \mathcal{D}^L}(l_{ce}(P_t^R, y) + l_{ce}(P_t^T, y)). \tag{1}$$

Here $(I_t^R, I_t^T)$ is a Query pair in the labeled set and $y$ is the corresponding ground-truth map. $l_{ce}$ denotes the cross-entropy loss function.

**Cross-collaborative Consistency Learning.** The key challenge in semi-supervised MVSS lies in how to mine effective supervision from *unlabeled* RGB-Thermal videos to complement label-guided training. In semi-supervised RSS, consistency regularization has achieved notable success, benefiting from perturbation-invariant training to enforce consistent predictions across various perturbations of unlabeled RGB images at different processing levels—input [22], feature [23, 24], or network [25].

This success motivates us to consider applying consistency regularization to RGB-thermal videos. To explore this, a simple yet effective Cross-collaborative Consistency Learning (C3L) module is devised, which leverages the unique properties of RGB-Thermal videos to perform perturbation-invariant training. Our intuition is that: RGB and thermal images inherently capture the same scene from distinct sensory perspectives, *i.e.,* visible light and thermal infrared, which provide innate input perturbations; further, processing these multimodal data through two parallel networks with distinct parameters introduces valuable feature-level and network-level perturbations. With these insights, our C3L is devised to apply mutual pseudo supervision between RGB and thermal streams to effectively utilize unlabeled RGB-Thermal frame pairs.

Specifically, we first compute a pair of one-hot pseudo-labels from the initial probabilistic segmentation predictions, $P_i^R$ and $P_i^T$, using the *argmax* function: $Y_i^R, Y_i^T = argmax(P_i^R), argmax(P_i^T)$. Then, the pseudo-labels are exploited to provide pseudo supervisions to the other stream, defined as:

$$\mathcal{L}_{c3l} = \mathbb{E}_{(I_i^R, I_i^T) \in \mathcal{D}^U \bigcup \mathcal{D}^L} (l_{ce}(P_i^R, Y_i^T) + l_{ce}(P_i^T, Y_i^R)). \tag{2}$$

Ideally, the C3L loss is expected to enhance the robustness of the model by fostering cross-modal consistency between unlabeled RGB-Thermal pairs. However, our experiments indicate a decrease in segmentation performance with this approach, as discussed in Sec. 5.4. We conjecture this decrease is primarily due to the inherent limitations of particular sensors, where the pseudo label generated from either RGB or thermal images alone might be incomplete, causing confused training and error accumulation. To counteract this, we need to consider effective cross-modal collaboration in the C3L framework. In terms of the design of cross-modal collaboration, a lot of fusion strategies [48, 49, 50, 13, 15] have been proposed in the MSS field as discussed in related works. Therefore, our work does not claim a new fusion strategy, but introduces the insights to highlight the importance of cross-modal collaboration for semi-supervised consistency regularization in MVSS. In C3L, we employ the Cross-modal Fusion block (CMF)[2] from [13] in our implementation and discuss other design choices in the supplements. By introducing cross-modal collaboration, the pseudo labels are refined by engaging the complementary information from the alternate stream, leading to significant improved performance.

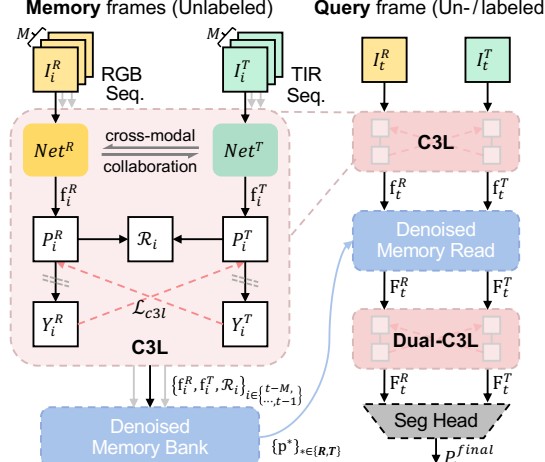

Figure 4: Overview of proposed method. For simplicity, the supervised losses are omitted. The C3L loss $\mathcal{L}_{c3l}$ (Eq. 2) aims to learn from unlabeled RGB-Thermal pairs. The DMR is responsible for integrating temporal information from the denoised memory bank to update query features. A dual-C3L loss (Eq. 7) is further applied to regularize updated query features. Finally, a segmentation head predicts the final mask $P_t^{final}$. The dotted \\ means stop gradient.

**Denoised Memory Read.** Next we consider how to integrate temporal information from past video frames. Among solutions in the related VSS and MVSS fields, attention-based memory read methods [21, 75, 17] have yield promising results, which selectively retrieve information from past (Memory) frames for improving current (Query) frame segmentation. In semi-supervised MVSS, due to the absence of ground-truth supervisions for past frames, Memory features are prone to be unreliable.

To deal with this issue, we further introduce a reliability estimation strategy, which can be easily integrated into existing temporal aggregation modules [21, 75] to mitigate potential noise. Here we utilize prototypical memory read [21] for efficient temporal aggregation. Our intuition is that reliable RGB and thermal features tend to yield consistent predictions. Conversely, discrepancies in these predictions can, to a certain extent, suggest potential unreliability. To quantify this, we design a normalized bidirectional Kullback–Leibler (KL) divergence function to estimate the pixel-wise

---

[2]CMF [13] is built on the gating mechanism that enables the model to emphasize useful features in one modality and compensate its missing information from the other.

reliability map as:

$$\mathcal{R}_i = 1 - \frac{1}{2}\big(\mathcal{N}(\sum_{c\in\mathcal{C}} P_i^R(c)\log\frac{P_i^R(c)}{P_i^T(c)}) + \mathcal{N}(\sum_{c\in\mathcal{C}} P_i^T(c)\log\frac{P_i^T(c)}{P_i^R(c)})\big). \tag{3}$$

Here, the KL divergence function is performed pixel-wisely (we omit the pixel scripts for simplicity); $\mathcal{N}(\cdot)$ is a min-max normalization function performed spatially to normalize divergence values to the range of 0 to 1. The final reliability map $\mathcal{R}_i$ is with the dimension of $H \times W \times 1$.

To efficiently store reliable memory features with minimal memory usage, we then establish a denoised prototype-based memory bank [21]. Concretely, for each memory feature $\mathbf{f}_i^* \in \mathbb{R}^{H\times W\times D}$ extracted from $Net^R$ and $Net^T$, where $* \in \{R, T\}$ indicates the image modality and $D$ is the channel dimension, we generate $C$ denoised class-level prototype features by spatially aggregating denoised features belonging to each category:

$$\mathbf{p}_i^* = \mathcal{G}(\mathbf{f}_i^* \times \mathcal{R}_i, Y_i^*) \in \mathbb{R}^{C\times D}, \tag{4}$$

Here $\times$ means pixel-wise multiplication to down-weight unreliable memory features based on the reliability map $\mathcal{R}_i$. $\mathcal{G}$ is the aggregation operation, which spatially averages the features belonging to each class based on pseudo-label $Y_i$. With this process, we obtain a condensed and denoised prototype-based memory bank $\{\mathbf{p}^* \in \mathbb{R}^{MC\times D}\}_{*\in\{R,T\}}$.

Subsequently, we use the attention mechanism to selectively retrieve relevant semantic information from the denoised memory bank, thereby refining query features. Taking RGB query feature $\mathbf{f}_t^R \in \mathbb{R}^{H\times W\times D}$ as an example, the updated RGB query feature $\mathbf{F}_t^R$ is derived as follows:

$$\mathbf{w}^* = Softmax(\bar{\mathbf{f}}_t^R \otimes transpose(\bar{\mathbf{p}}^*)), * \in \{R, T\}, \tag{5}$$
$$\mathbf{F}_t^R = \phi([\mathbf{w}^R\mathbf{p}^R, \mathbf{w}^T\mathbf{p}^T, \mathbf{f}_t^R]). \tag{6}$$

Here, $\bar{\mathbf{f}}_t^R$ and $\bar{\mathbf{p}}^*$ indicate $L_2$ normalized features, $\otimes$ denotes matrix multiplication, $[\cdot, \cdot, \cdot]$ means feature concatenation, and $\phi(\cdot)$ is a convolutional operation to adjust channel size.

The denoised memory read module finally outputs two enhanced query features $\mathbf{F}_t^R, \mathbf{F}_t^T \in \mathbb{R}^{H\times W\times D}$, which are enriched with useful denoised temporal contexts from unlabeled past frames.

**Dual-C3L.** In order to make full use of the unlabeled data and to further regularize the *memory-updated features*, we add C3L loss on the updated query features as well, called Dual-C3L loss:

$$\hat{\mathcal{L}}_{c3l} = \mathbb{E}_{(I_t^R, I_t^T)\in\mathcal{D}^U\bigcup\mathcal{D}^L}(l_{ce}(\hat{P}_t^R, \hat{Y}_t^T) + l_{ce}(\hat{P}_t^T, \hat{Y}_t^R)), \tag{7}$$

where $\{\hat{P}_t^R, \hat{P}_t^T\}$ are updated predictions inferred from updated query features $\{\mathbf{F}_t^R, \mathbf{F}_t^T\}$, and $\{\hat{Y}_t^R, \hat{Y}_t^T\}$ are corresponding pseudo labels. Accordingly, for the labeled query pairs, an additional supervision loss, $\hat{\mathcal{L}}_{sup}$, is also applied on the updated predictions, similar to Eq. 1.

**Final Prediction and Training Objective.** To infer the final output, the updated query features $\{\mathbf{F}_t^R, \mathbf{F}_t^T\}$ are concatenated together, followed by a $3 \times 3$ convolutional layer as segmentation head to predict the final mask $P_t^{final}$. A supervised cross-entropy loss is also applied to $P_t^{final}$, as:

$$\mathcal{L}_{sup}^{final} = \mathbb{E}_{(I_t^R, I_t^T, y)\in\mathcal{D}^L} l_{ce}(P_t^{final}, y). \tag{8}$$

The overall training objective of the proposed SemiMV framework is thus defined as:

$$\mathcal{L}_{total} = \mathcal{L}_{sup} + \hat{\mathcal{L}}_{sup} + \mathcal{L}_{sup}^{final} + \lambda(\mathcal{L}_{c3l} + \hat{\mathcal{L}}_{c3l}), \tag{9}$$

where $\lambda$ is the trade-off weight to balance the supervised losses and pseudo losses from C3L.

## 5 Experiments

### 5.1 Datasets and Evaluation Metric

Our experiments are conducted on both MVSeg and MVUAV datasets. *MVSeg* [21] is a street scene dataset with 26 semantic classes. It consists of 452, 84, and 202 videos in its training, validation, and testing subsets, respectively. Annotations are provided for every 15 frames, resulting in 2,241 training, 378 validation, and 926 testing semantic masks. *MVUAV* provides bird's-eye view scenes with 36 semantic classes. The dataset splits are mentioned in Sec. 3. We follow the partition protocols of [25] and divide the whole training set via randomly sub-sampling 1/2, 1/4, 1/8, and 1/16 training videos as the labeled set, and treat the remaining videos as the unlabeled set. The numbers of annotated videos/frames for each training partition, denoted as (#videos, #frames), are included in Table 2 and Table 3. By convention [21], we adopt the mean Intersection over Union (mIoU) for evaluation.

## 5.2 Implementation Details

The model is implemented on the Pytorch and trained using two NVIDIA A100 GPUs. To be consistent with previous work [25, 21], we adopt DeepLabv3+ [34] with ResNet50 as backbone, for both RGB and thermal streams. For the RGB stream, we initialize the network parameters using weights pretrained on ImageNet [92]. For the thermal stream, we randomly initialize the network parameters, and generate 3-channel thermal images as inputs by repeating the 1-channel thermal images. All training images are resized to $320 \times 480$. The model is optimized by Adam with batch size of 2, and the learning rate is 2e-4 which is annealed following the poly LR policy. The network converges around 200 epochs. We follow [21] to adopt three reference frames ($M = 3$) at a sample rate of 3 as Memory. $\lambda$ is empirically set as 1. The network training involves two-stages: the first stage is backbone warming-up trained with only annotated query frames (50 epochs), and the second stage is main-training of SemiMV trained with all videos (150 epochs). During testing, the SemiMV processes each frame sequentially, inferring results in just 19.1 ms per frame.

## 5.3 Comparison with the State-of-the-Arts

Since our SemiMV is the first work to address semi-supervised MVSS, to provide a reference level, we reimplement some related approaches in *semi-supervised RSS* (MT [22], CCT [23], CPS [25], and UniMatch [24]), *semi-supervised VSS* (IFR [80]), *VSS* (Accel [77]), and *MVSS* (MVNet [21]) fields, using their official codes. These models use ResNet50 [93] as feature extractor.

**Results on MVSeg Dataset.** Table 2 lists the segmentation results on the MVSeg dataset. We show the fully-supervised performance achieved by training only on the labeled data using different modalities, in the first row of each block. We observe that the semi-supervised models consistently surpass their supervised baselines across all data partitions, highlighting the significance of semi-supervised learning in semantic segmentation. In particular, our SemiMV improves the SupOnly (RGBT) baseline by large margins, *i.e.*, +2.22%, +5.67%, +6.16%, +5.98%, under 1/16, 1/8, 1/4, 1/2 partition protocols, respectively. This demonstrates the effectiveness of our SemiMV in engaging unlabeled multispectral videos to enhance the generalization capabilities of segmentation models. Moreover, compared to MVNet [21], which utilizes sparsely labeled videos alone, our SemiMV consistently performs better. This is attributed to the simultaneous usage of both unlabeled past frames and extensive unlabeled multispectral videos in our SemiMV framework.

Table 2: Quantitative evaluation on the MVSeg dataset. SupOnly stands for the model trained on the labeled data.

| Method | 1/16 (26,140) | 1/8 (54,282) | 1/4 (111,561) | 1/2 (228,1119) |
|---|---|---|---|---|
| SupOnly (RGB) | 21.95 | 27.09 | 35.79 | 42.37 |
| MT [22] | 23.39 | 29.45 | 38.75 | 44.51 |
| CCT [23] | 23.81 | 29.66 | 39.04 | 44.89 |
| CPS [25] | 23.88 | 30.05 | 39.27 | 45.34 |
| UniMatch [24] | 24.73 | 30.47 | 39.39 | 45.42 |
| Accel [77] (Video) | 23.16 | 28.41 | 37.31 | 43.75 |
| IFR [80] | 24.79 | 30.97 | 40.69 | 46.21 |
| SupOnly (RGBT) | 23.26 | 28.45 | 36.88 | 43.75 |
| MVNet [21] | 24.70 | 30.32 | 39.89 | 46.08 |
| **SemiMV (Ours)** | 25.48 | 34.12 | 43.04 | 49.73 |

**Results on MVUAV Dataset.** Table 3 reports the comparison results on the MVUAV dataset. Benefiting from the integration of rich multispectral video information and the powerful capability of semi-supervised learning to utilize unlabeled data, our SemiMV achieves the highest performance. The improvement of our SemiMV over the supervised baseline SupOnly (RGBT) are +3.82%, +5.16%, +5.21%, +4.63% under 1/16, 1/8, 1/4, 1/2 partition protocols, respectively. Overall, our architecture can adapt to various settings and scenarios by achieving superior performance on both MVSeg and MVUAV datasets. This showcases the robustness of our SemiMV framework.

Table 3: Quantitative evaluation on the MVUAV dataset.

| Method | 1/16 (23,91) | 1/8 (40,184) | 1/4 (70,365) | 1/2 (141,732) |
|---|---|---|---|---|
| SupOnly (RGB) | 10.09 | 13.47 | 20.07 | 26.25 |
| MT [22] | 11.33 | 15.89 | 23.02 | 27.83 |
| CCT [23] | 11.75 | 16.11 | 23.72 | 28.71 |
| CPS [25] | 12.55 | 16.70 | 24.01 | 29.09 |
| UniMatch [24] | 13.36 | 17.21 | 24.10 | 29.21 |
| Accel [77] (Video) | 11.23 | 14.69 | 21.45 | 27.70 |
| IFR [80] | 13.11 | 17.03 | 24.91 | 29.87 |
| SupOnly (RGBT) | 11.28 | 14.88 | 21.31 | 27.60 |
| MVNet [21] | 13.07 | 16.86 | 23.36 | 29.77 |
| **SemiMV (Ours)** | 15.10 | 20.04 | 26.52 | 32.23 |

**Visual Comparison.** Fig. 5 illustrates the qualitative results of different methods on the MVSeg dataset under 1/4 partition protocol. As seen, the MVNet and our SemiMV reconciling both multispec-

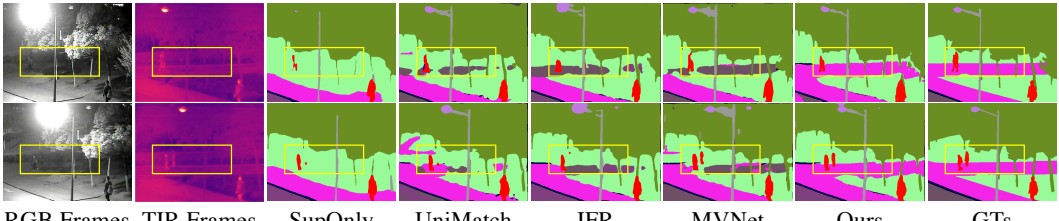

| RGB Frames | TIR Frames | SupOnly | UniMatch | IFR | MVNet | Ours | GTs |

Figure 5: Qualitative results on MVSeg dataset. We highlight the details with the yellow boxes.

tral and temporal contexts can better identify objects in low-light places, for example, the pedestrians within the yellow boxes, while other methods struggle. Moreover, the results of our method are closer to the ground truths than MVNet, attributing to the combined benefits of semi-supervised learning and multispectral video data. *More results are provided in the supplementary materials.*

## 5.4 Ablation Studies

In this section, we conduct ablation studies on the MVSeg dataset under 1/4 partition setup.

**Effect of each component.** In Table 4, we evaluate the performance improvements achieved by systematically integrating key components into our framework. Initially, we compare two supervised baselines: RGB *vs.* RGB-Thermal, each trained with labeled RGB frames or RGB-Thermal frame pairs, respectively. It is observed that incorporating a thermal infrared branch brings a notable performance gain of 1.09%, verifying the value of multispectral information in semantic segmentation. As the proposed C3L and DMR are gradually incorporated, increased performance is consistently observed. In particular, C3L enhances the supervised baseline by a remarkable amount of +3.85% (36.88% → 40.73%), demonstrating its efficacy in harnessing unlabeled RGB-Thermal frame pairs. The addition of our DMR further boosts performance by +1.66% mIoU, benefiting from the exploitation of denoised and context-rich unlabeled past frames. Notably, our Dual-C3L loss added on updated query features further elevates the mIoU score to 43.04%, amplifying the combined strengths of our C3L and DMR modules in leveraging unlabeled multispectral videos for Semi-MVSS.

Table 4: Ablation study of the proposed SemiMV framework.

| Methods | Information | | | | | mIoU |
| | RGB | Thermal | Labeled | Unlabeled | Video | |
|---|---|---|---|---|---|---|
| RGB | ✓ | | ✓ | | | 35.79 |
| RGB-Thermal | ✓ | ✓ | ✓ | | | 36.88 |
| +C3L | ✓ | ✓ | ✓ | ✓ | | 40.73 |
| +DMR | ✓ | ✓ | ✓ | ✓ | ✓ | 42.39 |
| +Dual-C3L | ✓ | ✓ | ✓ | ✓ | ✓ | 43.04 |

**Analysis on C3L.** We then delve into the designs of C3L in Table 5. To effectively utilize unlabeled RGB-Thermal frame pairs, our C3L leverages cross pseudo labels as supervision signals and introduces cross-modal collaboration to enable the generation of reliable pseudo labels. To verify their significance, we develop two variants, $C3L_1^-$ w/o cross supervision and $C3L_2^-$ w/o cross collaboration. In $C3L_1^-$, the mean-teacher output of each stream is used to generate pseudo label for itself without using cross supervisions. This leads to a reduction of 1.61% mIoU compared to the standard C3L. We conjecture this drop is due to the accumulation of self-errors, whereas our C3L, with its cross supervision, has the ability to mutually correct potential errors, thereby enhancing overall accuracy. Moreover, it is worth noting that the $C3L_2^-$ variant, which applies cross pseudo supervision without the benefit of cross-modal collaboration, leads to a deterioration in result. This indicates that cross-modal collaboration is critical and indispensable for the effective functioning of our Semi-MVSS framework.

Table 5: Ablation analysis of our C3L module.

| Ablation analysis on C3L | Labeled | Unlabeled | mIoU |
|---|---|---|---|
| Baseline | ✓ | | 36.88 |
| $C3L_1^-$ *w/o cross supervision* | ✓ | ✓ | 39.12 |
| $C3L_2^-$ *w/o cross collaboration* | ✓ | ✓ | 36.67 |
| C3L | ✓ | ✓ | 40.73 |

**Analysis on DMR.** Table 6 presents our DMR-Proto and an alternative module - DMR-STM as in [75]. DMR-STM performs an all-to-all attention mechanism for matching between query and memory frames. Our results reveal that, 1) engaging temporal contexts from unlabeled past frames is indeed useful, as both DMR modules yield increased mIoU scores; 2) the proposed reliability estimation strategy

Table 6: Ablation analysis of our DMR module.

| Methods | w/o temporal | DMR-STM | | DMR-Proto | |
| | | w/o denoised | with denoised | w/o denoised | with denoised |
|---|---|---|---|---|---|
| mIoU | 40.73 | 41.47 | 42.25 | 41.85 | 42.39 |
| Δ | - | (+0.74) | (+1.52) | (+1.12) | (+1.66) |

effectively filters out unreliable features, enhancing the performance of both DMR variants. Considering computational efficiency, DMR-Proto is used for temporal integration in our SemiMV network. *Additional model analyses are provided in the supplementary materials.*

## 6  Conclusion

This study introduces MVUAV, a new multispectral video semantic segmentation dataset obtained through UAVs from oblique bird's-eye viewpoint. The dataset, accompanied with precise semantic annotations, serves as a complementary resource to the MVSeg dataset, offering a broader perspective for evaluating MVSS models. Additionally, this paper pioneers the development of SemiMV, the first semi-supervised MVSS framework tailored to utilize both labeled and unlabeled multispectral videos effectively. Comprehensive empirical results affirm the effectiveness of our approach and highlight the promising potential of semi-supervised leaning in MVSS.

**Broader Impacts.** The proposed MVUAV dataset offers significant value in enhancing the performance of semantic segmentation and propelling further research in the field of MVSS. We envisage that the most proximate impacts of this dataset will be positive, providing a valuable asset for researchers and developers in the field. Our semi-supervised MVSS method makes robust scene parsing available without intensive human annotation efforts, which saves a lot of costs. Note that the proposed MVUAV dataset is strictly prohibited from being used to identify or invade the privacy of any individual and is made available solely for academic purposes. In addition, we discuss potential limitations with some feasible solutions in the supplements.

**Reproducibility Statement.** Our source code and dataset, along with easy-to-follow instructions, are publicly available on our project website.

## 7  Acknowledgements

This work was partially supported by the the Alberta Innovates CASBE - NSERC Alliance and the NSERC Discovery (RGPIN-2019-04575) grants, and Samsung AI Center-Mountain View.

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
