# OpenReview forum: "Unleashing Multispectral Video's Potential in Semantic Segmentation: A Semi-supervised Viewpoint and New UAV-View Benchmark"
_NeurIPS.cc/2024/Conference — NeurIPS 2024 poster_

### Official Review · Reviewer_Mx7S · 2024-07-01

**Soundness:** 2
**Presentation:** 2
**Contribution:** 2
**Rating:** 6
**Confidence:** 4

**Summary:**

The paper proposes advancements in multispectral video semantic segmentation (MVSS) through two key contributions: the creation of a new benchmark dataset, MVUAV, captured via UAVs, and the development of SemiMV, a semi-supervised learning baseline designed to optimize sparse annotations using Cross-collaborative Consistency Learning (C3L).

**Strengths:**

1. The MVUAV dataset introduces an oblique bird’s-eye view for multispectral video semantic segmentation, providing rich and diverse data that include a wide array of lighting conditions and over 30 semantic categories.

2. The SemiMV framework uses semi-supervised learning for MVSS tasks, employing a Cross-collaborative Consistency Learning (C3L) module and a denoised temporal aggregation strategy, offering a solution to utilize sparse annotations and unlabeled data.

3. The paper's empirical evaluations confirm that the SemiMV baseline enhances the multispectral video semantic segmentation.

**Weaknesses:**

1. The paper presents a new dataset for semantic segmentation from a high-altitude perspective, yet lacks a detailed comparison with existing aerial-view datasets [1, 2, 3, 4] which is necessary to establish the dataset's relevance and uniqueness.

2. The paper fails to articulate the motivation and significance behind the introduction of the UAV-View dataset, leaving readers uncertain about the necessity and potential contributions of this new dataset to the field.

3. There is an absence of a thorough analysis of limitations and broader impacts, which is a concern as it may not meet submission requirements that typically expect such discussions to understand the full implications and potential drawbacks of the research.

4. The paper does not adequately address privacy concerns regarding the UAV-view multimodal dataset, such as the potential capture of private information like pedestrians and storefronts, and fails to clarify whether appropriate privacy measures are in place or if there is official and public approval for data collection in the relevant regions.

[1] Vision Meets Drones: A Challenge

[2] The Unmanned Aerial Vehicle Benchmark: Object Detection and Tracking

[3] Ultra-High Resolution Segmentation with Ultra-Rich Context: A Novel Benchmark

[4]  LoveDA: A Remote Sensing Land-Cover Dataset for Domain Adaptive Semantic Segmentation

**Questions:**

Please see the weakness

**Limitations:**

Please see the weakness

---

> ### Author Rebuttal · Authors · 2024-08-06
>
> Dear Reviewer Mx7S, we sincerely appreciate the time and effort you spent reviewing our paper and your positive feedback. Your comments are insightful, and we look forward to addressing each of your concerns point-by-point.
>
> ---
>
> ***W1**: "The paper presents a new dataset for semantic segmentation from a high-altitude perspective, yet lacks a detailed comparison with existing aerial-view datasets [1, 2, 3, 4] which is necessary to establish the dataset's relevance and uniqueness."*
> **Response**: Thanks for your valuable suggestion to include a detailed comparison with the mentioned aerial-view datasets [1, 2, 3, 4], which will make our work more comprehensive. In comparison, **VisDrone2018 [1] and UAVDT [2] are two large-scale datasets designed primarily for object detection and tracking tasks** in UAV-view RGB videos and/or images, providing bounding box annotations for target objects. In contrast, **our MVUAV dataset is focused on the semantic segmentation task** in UAV-view RGB-thermal videos, offering dense pixel-wise semantic annotations. **URUR [3] and LoveDA [4] are two high-resolution segmentation datasets** collected by high-quality satellite or Spaceborne images. **The key advantage of our MVUAV dataset compared to URUR and LoveDA is the inclusion of complementary multispectral (RGB-thermal) videos.** This feature aids in detecting target objects at nighttime or in adverse lighting conditions, thereby enhancing low-light vision capabilities. **In the PDF file of the general response, we present the detailed statistics of these datasets [1, 2, 3, 4] and our MVUAV dataset in Table C.** Following your suggestion, we will include these discussions and detailed analyses in our paper.
>
> ---
>
> ***W2**: "The paper fails to articulate the motivation and significance behind the introduction of the UAV-View dataset, leaving readers uncertain about the necessity and potential contributions of this new dataset to the field."*
> **Response**: We greatly appreciate the opportunity to better articulate the motivation and significance of our UAV-view multispectral video semantic segmentation dataset (MVUAV). **The significance of MVUAV can be illustrated through the following points: (1) Importance of UAV-View Characteristics:** UAV-view data provide a broader, holistic perspective free from the constraints of ground-level capture. This characteristic has demonstrated to be advantageous for many applications in computer vision, such as detection [45], tracking [56], and segmentation [33].
> **(2) Capability of Low-Light Vision:** Compared to existing UAV-view RGB segmentation datasets like UAVid [33], our dataset offers a unique combination of RGB and thermal infrared videos, enhancing low-light vision capabilities that existing works do not cover.
> **(3) Advancement of MVSS Task:** From a complementary perspective, our MVUAV dataset offers a distinct bird's-eye viewpoint that complements existing ground-level datasets like MVSeg. The presence of both datasets enriches the diversity of perspectives available in the field of MVSS, enabling more comprehensive analysis and validation of algorithms across various scenarios. This is particularly advantageous for applications requiring comprehensive coverage in challenging conditions, such as aerial nighttime search and rescue, sea patrols, firefighting response support, traffic management, and UAV delivery services. Thanks for your valuable suggestion. **We will add this motivation and discussion in our final paper to make it more clear.**
>
> ---
>
> ***W3**: "There is an absence of a thorough analysis of limitations and broader impacts, which is a concern as it may not meet submission requirements that typically expect such discussions to understand the full implications and potential drawbacks of the research."*
> **Response**: Thanks for your valuable feedback. We respectfully remind the reviewer that a detailed discussion on limitations and broader impacts is included in Appendix A.6. We apologize for any confusion caused due to content layout. In the revised paper, we will ensure that proper reference is added to the main text to guide readers to our discussion.
>
> ---
>
> ***W4**: "The paper does not adequately address privacy concerns regarding the UAV-view multimodal dataset, such as the potential capture of private information like pedestrians and storefronts, and fails to clarify whether appropriate privacy measures are in place or if there is official and public approval for data collection in the relevant regions."*
> **Response**: Thank you for raising the important issue of privacy concerns. In our MVUAV dataset, we provide rich pixel-level semantic segmentation annotations for selected multispectral UAV videos from VTUAV [56] and we do not create new source data. We have obtained official approvals from the original authors and institutions to further process and re-annotate their videos with semantic labels to advance the development of the MVSS field.
> **Following your suggestions, we will release de-identified videos using defacing and storefront detection tools to mitigate potential privacy issues, and we will also claim that we prohibit people from using our MVUAV in any manner to identify or invade the privacy of any person**. Additionally, **our MVUAV dataset will be made freely available solely for academic purposes.**
>
> ---
>
> ***Ethics Review***: Please kindly refer to our responses to W3 and W4.
>
> ---
>
> Once again, thank you for your valuable time and comments for enhancing the quality of our paper. We hope our response can address your concerns. If there are any further questions, please feel free to share, and we are happy to address them.

---

> > ### Comment · Reviewer_Mx7S · 2024-08-09
> >
> > Thanks for addressing my concerns in the rebuttal. I raise my score to weak accept. Please make the necessary changes and references as noted in the rebuttal.

---

> > > ### Author Response · Authors · 2024-08-09
> > >
> > > Dear Reviewer Mx7S, we are delighted to see that your questions and concerns have been addressed. We will carefully revise our paper, incorporating the necessary changes and references based on your suggestions. Additionally, our dataset, source code, and project website with easy-to-follow guidelines, will be made publicly available to the research community. Thank you once again for your valuable feedback and efforts in helping us improve our paper.

---

### Official Review · Reviewer_XBRU · 2024-07-10

**Soundness:** 2
**Presentation:** 1
**Contribution:** 2
**Rating:** 5
**Confidence:** 5

**Summary:**

This paper introduces a new multi-spectral aerial-view semantic segmentation dataset called MVUAV, which consists of 413 video sequences and 53K frames with sparsely annotated pixel-level segmentation labels. To provide a way of using this sparsely annotated dataset, it also introduces a new semi-supervised semantic segmentation method that takes RGB and thermal image pairs as input. The newly designed key component in this method is cross-collaborative consistent learning, which uses the segmentation predictions of one modal network as pseudo-labels for training another modal network. This paper demonstrates the effectiveness of this method with better performance than other baselines on a multi-spectral semi-supervised semantic segmentation task on two benchmark datasets including MVUAV.

**Strengths:**

1. MVUAV, a new dataset, is very valuable in the field of multi-spectral semantic segmentation. I admit that the aerial-view has more and smaller semantics than the ground-view (called eye-level view in the paper) due to its wider viewing angle, so labeling takes significantly more time.

2. SemiMV, a new method introduced in the paper, showed better performance than other baselines on two benchmark datasets evaluating multi-spectral semantic segmentation tasks.

**Weaknesses:**

1. Immature presentation
--

Overall, the presentation in the paper is not mature for publication.

a. Section 4.2 is not easy to follow due to missing description of some terminology.

- What is the memory feature  the memory feature $f^*_i$ and how is it obtained in eq 4?

- In eq 5, the dimensions of $f_t$ and $transpose(p)$ are H$\times$W$\times$1 and MC$\times$D, respectively. Here, the dimension of $f_t$ is guessed from eq 4, in which $f$ is element-wise multiplied with $\mathcal{R}$ of the dimension H$\times$W$\times$1. How can the matrix multiplication be applied to these two variables?

b. Figures do not deliver any crucial and complicate information.

- Figure 2 is not necessary to understand what it conveys.

- Figure 4 is not referred in the contents, so it is unclear what this figure exists to support.


2. Insufficient contribution
--

Contribution is not sufficient to meet NeurIPS standards.

a. As this paper also mentioned, the proposed network does not have any novel component. C3L is very simple and not new as already used in various previous works (works in Ln 187).

b. Dataset is somewhat novel as it contains aerial-view images. However, no experiment is provided to demonstrate why acquiring this characteristics in dataset is important.

3. Insufficient experiments
--

a. All baseline methods compared in table 2 and 3 are developed not for the purpose of semi-supervised learning. Without comparing other semi-supervised learning method, it is difficult to figure out the effectiveness of the proposed method.

b. Ablation studies for several parameters which are crucial in the method are needed, e.g., M and lambda.

**Questions:**

Please address weaknesses I pointed out.

**Limitations:**

.Limitation and potential societal impact are not mentioned in the main manuscript but included in supplementary material.

---

> ### Author Rebuttal · Authors · 2024-08-06
>
> Dear Reviewer XBRU, thank you for your recognition that our MVUAV dataset is valuable and our SemiMV method shows better performance. We hope to address each of your questions point-by-point and clarify some misunderstandings.
>
> ---
>
> ***Q1-a**: "(**a1**) What is the memory feature* $f_i^*$, ...*" and "(**a2**) In eq 5, ... How can the matrix multiplication be applied to these two variables?"*
> **Response to Q1-a**:
> (**a1:**) $f_i^*$ **represents the decoded feature extracted from segmentation nets (i.e., $Net^R$ and $Net^T$ in Fig. 4) at the $i_{th}$ past frame** ($i \in [t-M, \cdots, t-1]$), where $* \in\\{R,T\\}$ indicates the image modality and $M$ is the number of past frames stored in memory for temporal utilization. Here we can adopt common segmentation networks such as DeepLabv3+ [6] and SegFormer [50] to obtain $f_i^*$ $\in \mathbb{R}^{H\times W\times D}$. A summary of notation definitions (including $f_i^*$) was presented in Table 7. **In Eq. 4**,  $f_i^*$ **is used to generate denoised prototype feature** $p_i^* \in \mathbb{R}^{C\times D}$.
>
> (**a2:**) Similar to $f_i^*$, $f_t^*$ **has the dimension of $H\times W\times D$, not $H\times W\times 1$.** Thus, the matrix multiplication can be directly applied on their two $L_2$ normalized variables, $\bar{f}_t^R \in \mathbb{R}^{H\times W\times D}$ and $transpose(\bar{p}^*) \in \mathbb{R}^ {D\times MC}$, thereby obtaining the attention weight $\textbf{w} \in \mathbb{R}^ {H\times W\times MC}$ with softmax function in Eq. 5.
>
> **For better understanding, we provide a diagram (Figure A) in the PDF file of the general response**, illustrating the feature transformation process and corresponding dimension changes of Eqs. 4 & 5 & 6. If you have any other questions, please let us know.
>
> ---
>
> ***Q1-b1**: "Figure 2 is not necessary ..."*
> **Response**: We respectfully believe Figure 2 is valuable as our work is the first to address the MVSS task from a semi-supervised perspective. It helps readers quickly understand the use of training data in this context, especially for those who may be unfamiliar with semi-supervised MVSS.
>
> ---
>
> ***Q1-b2**: "Figure 4 is not referred in the contents ..."*
> **Response**: Figure 4 is referred to at the beginning of Sec. 4.2 (line 173). It depicts the overall architecture of our proposed SemiMV.
>
> ---
>
> ***Q2-a**: "As this paper also mentioned, the proposed network does not have any novel component ..."*
> **Response**: **We respectfully believe that our proposed method is valuable and introduces new insights compared to previous works, including: 1) Cross-collaborative Consistency Learning:** Previous works [7,36,43,53] are specifically designed for single-modality RGB images, while our SemiMV excels in processing unlabeled multispectral videos by engaging their dual-perspective characteristic and cross-modal collaboration. Our experiments (lines 359-374) show that directly adapting [7,43] to semi-supervised MVSS is ineffective because they overlook the importance of cross pseudo-supervision and cross-modal collaboration. **2) Denoised temporal strategy:** We introduce a pixel-wise reliability map based on the learned cross-modal consistency to guide the temporal fusion process and mitigate noise. This addresses the noisy memory feature issue not covered by previous works [22,35,37]. **3) Extensive experiments:** Tables 4-6 & 12 verify the superiority of our SemiMV through investigating various design choices for semi-supervised MVSS. Tables 10 & 11 show that our method performs well with different backbones (CNN and transformer) and achieves consistent performance improvements when integrated with existing semi-supervised schemes. These results indicate that our SemiMV can serve as a scalable baseline for future work. **We will make our source code publicly available with easy-to-follow guidelines.**
>
> ---
>
> ***Q2-b**: "Dataset is somewhat novel as it contains aerial-view ..."*
> **Response**: Thank you for recognizing the novelty of our dataset. Regarding why acquiring these characteristics in the dataset is important, please kindly refer to our response to Reviewer Mx7S's W2, which articulates the significance of MVUAV through three aspects. Note that we are not claiming that UAV-view data are superior to ground-level counterparts. We aim to highlight their unique benefits. Our MVUAV dataset offers a distinct bird's-eye view that complements the existing ground-level MVSeg, thereby enriching the diversity of perspectives in MVSS. Given the extensive literature on UAV-view research [33,45,56] and the aforementioned advantages, we believe this perspective is beneficial for the MVSS task. We hope this clarifies your concern.
>
> ---
>
> ***Q3-a**: "All baseline methods compared in table 2 and 3 are developed not for the purpose of semi-supervised learning. ..."*
> **Response**: We respectfully remind the reviewer that MT, CCT, CPS, UniMatch, and IFR in Tables 2 and 3 are specifically developed for semi-supervised learning, as mentioned in lines 293-296. Additionally, Tables 4-6 and 9-14 present extensive ablation studies, various design choices, different backbones, and integration with other semi-supervised schemes, demonstrating the scalability and effectiveness of our method.
>
> ---
>
> ***Q3-b**: "Ablation studies for several parameters ... e.g., M and lambda.."*
> **Response**: Following your suggestion, we investigated the impact of $M$ and $\lambda$, **with results in Tables A & B of the one-page PDF**. Adding memory frames consistently improves mIoU scores, with a noticeable increase from  40.73% to 43.04% when $M=3$. Raising $M$ further beyond 3 gives marginal returns. Thus, we set $M=3$ for a better trade-off between accuracy and memory cost. For $\lambda$, we found that $\lambda=1$ balances supervised and pseudo losses effectively. These results will be included in our final paper.
>
> ---
>
> Thanks again for dedicating your valuable time and effort to our paper. If there are other questions, please let us know.

---

> > ### Comment · Reviewer_XBRU · 2024-08-13
> >
> > As the authors' rebuttal addresses most of my questions, I am raising my initial rating.

---

> > > ### Author Response · Authors · 2024-08-13
> > >
> > > Dear Reviewer XBRU, we sincerely appreciate your feedback and are delighted to see the raised score. We will carefully revise our paper according to your suggestions. Our dataset, source code, and project website will also be released to the public. Thank you once again for your valuable time and effort in strengthening our paper.

---

### Official Review · Reviewer_w3ua · 2024-07-12

**Soundness:** 3
**Presentation:** 3
**Contribution:** 4
**Rating:** 7
**Confidence:** 4

**Summary:**

The paper addresses multispectral video semantic segmentation (MVSS) and proposes a new semi-supervised learning approach. It introduces the SemiMV framework, which utilizes a Cross-collaborative Consistency Learning (C3L) module and denoised temporal aggregation strategy. Additionally, the paper establishes the MVUAV benchmark dataset, captured by UAVs, offering unique bird’s-eye views and various semantic categories.

**Strengths:**

S1: The paper presents a dataset that clearly must have required a lot of effort and resources to gather. It has the potential to contribute significantly to the community.


S2: The authors also provide a semi-supervised MVSS baseline, which demonstrates the practical application and effectiveness of their data.


S3: The dataset offers annotations and has advantages over existing datasets in terms of resolution, modality, annotation quality, and dataset size.

**Weaknesses:**

W1: Compared to UAVid, the resolution seems somewhat low, especially considering the UAV perspective where scenes are typically larger and require higher resolution for detailed analysis.

W2: The focus on UAV-captured data might limit the dataset’s applicability to ground-level or other perspectives, potentially reducing its versatility.

**Questions:**

Q1: The paper should address whether the RGB and thermal (TIR) images in the MVUAV dataset are precisely aligned. Some existing RGB-T datasets [1]  have issues with modality misalignment, which serveral previous research [2][3] has tried to address. Therefore I wonder if this dataset exist same misalignment issue.

Q2: The author should clarify the relationship between MVUAV and MVSeg. Is MVUAV merely a complementary perspective, or does it offer other different scene?

Q3: Given the sparse annotations, can this dataset be used for general segmentation tasks? Or if it is only suitable for semi-supervised  methods?


[1] Multispectral Pedestrian Detection: Benchmark Dataset and Baseline
[2] Weakly Aligned Cross-Modal Learning for Multispectral Pedestrian Detection
[3] Attentive Alignment Network for Multispectral Pedestrian Detection

**Limitations:**

The authors acknowledge that existing MVSS datasets, including MVUAV, are small due to high labeling costs, and while semi-supervised methods help, they don't fully meet real-world demands. They also mention that the SemiMV baseline and MVUAV dataset, though promising, face challenges such as small targets and scale variation, necessitating further research and enhancements.

---

> ### Author Rebuttal · Authors · 2024-08-06
>
> Dear Reviewer w3ua, we greatly appreciate the time and effort you have dedicated to providing constructive suggestions on ways to strengthen our paper. We are also grateful for the positive comments and recognition. Below, we make a point-by-point response to all the comments.
>
> ---
>
> ***W1**: "Compared to UAVid, the resolution seems somewhat low, ... ."*
> **Response**: Thank you for your constructive comment. We agree that higher resolution are often desirable for detailed analysis, especially from the UAV perspective. Compared to UAVid, **the resolution difference in our MVUAV arises from the inherent characteristics of thermal infrared cameras compared to RGB cameras.** While our MVUAV dataset has a relatively lower resolution than the RGB-based UAVid dataset, it uniquely **offers both RGB and thermal UAV videos, providing essential complementary information to enhance low-light vision**, which UAVid does not offer. Moreover, **compared to existing *RGB-thermal datasets*** such as image-only UAV dataset CART [23] and video dataset MVSeg [22], **our MVUAV features a relatively higher resolution** (1920×1080 in MVUAV vs. 960×600 in CART and 640×480 in MVSeg). For higher resolution needs, we could employ off-the-shelf super-resolution tools to enhance the image quality. We will include this discussion in our paper.
>
> ---
>
> ***W2**: "The focus on UAV-captured data might limit the dataset’s applicability to ground-level or other perspectives, ... ."*
> **Response**: Thanks for your valuable comment. We humbly believe that *UAV-view data and ground-level data have different characteristics and advantages, making them suited to different application scenarios.* From a complementary perspective, **our MVUAV dataset offers a distinct bird's-eye viewpoint that complements existing ground-level datasets** like MVSeg. The presence of both datasets could **enrich the diversity of perspectives available in the field of MVSS, enabling more comprehensive analysis and validation of algorithms** across various scenarios.
> Furthermore, **UAV-captured data provide a broader, holistic view, free from the constraints of ground-level capture.** This characteristic is **advantageous for applications that require comprehensive coverage in challenging conditions,** such as aerial nighttime search and rescue, sea patrols, firefighting response support, traffic management, and UAV delivery services.
>
> ---
>
> ***Q1**: "The paper should address whether the RGB and thermal (TIR) images in the MVUAV dataset are precisely aligned. ... ."*
> **Response**: Thank you for your insightful question. **We concur with the importance of well-aligned RGB-T pairs for segmentation tasks, and have proactively taken efforts to ensure the quality of our dataset. Specifically, this awareness has been taken into account during the dataset collection and preparation stages of both the sourced VTUAV dataset [56] and our MVUAV dataset.** In the VTUAV dataset, the authors manually identified corresponding feature points on both RGB and thermal images and calculated an affine transformation matrix from these points. Using this matrix, one image was warped to align with the other, and the common overlapping regions were extracted and resized to a consistent resolution while maintaining the aspect ratio. *This ensures that most frames are well-aligned. Additionally, we performed a visualization screening process* by overlaying thermal heat maps onto paired RGB images. This made it easier for our inspectors to verify alignment and allowed us to filter out low-quality samples (e.g., similar content, blurred, or misaligned images), *thus further enhancing the overall quality of the MVUAV dataset.* We will include these discussions and add related works in our final paper.
>
> ---
>
> ***Q2**: "The author should clarify the relationship between MVUAV and MVSeg. ... ."*
> **Response**: Thanks for your constructive suggestion for improving our paper.
> **Our MVUAV dataset not only offers a distinct bird's-eye viewpoint that complements existing ground-level MVSeg dataset, but also includes additional scenarios.** Thanks to the unique characteristics of UAVs, which provide a broader and more holistic view free from the constraints of ground-level capture, **MVUAV encompasses extra challenging scenes such as** rivers, boats, bridges, and playgrounds, as shown in Figure 3. It also covers a diverse set of 36 semantic classes. **Various visual scenes can be accessed on *our project website*.**
> We will discuss this in the revised paper.
>
> ---
>
> ***Q3**: "Given the sparse annotations, can this dataset be used for general segmentation tasks? ... ."*
> **Response**: Thanks for your question. **Yes, our dataset can be used for general fully-supervised segmentation tasks,** in addition to the semi-supervised setting. **For example,** researchers can directly utilize our labeled RGB-Thermal samples in the MVUAV dataset for multispectral (RGB-thermal) semantic segmentation (MSS) task. **In Table 9 of the Appendix, we provide comprehensive benchmarking results of various segmentation methods on the new MVUAV dataset under the fully-supervised setting.** *These results, along with our dataset and source code, will be made publicly available*. We hope this will support researchers in addressing their specific needs.
>
> ---
>
> Thanks again for your encouragement and insightful suggestions. If there are other questions, please let us know.

---

> > ### Comment · Reviewer_w3ua · 2024-08-12
> >
> > I appreciate your efforts in addressing my concerns in the rebuttal. Based on your responses, I am increasing my score to accept.

---

> > > ### Author Response · Authors · 2024-08-12
> > >
> > > Dear Reviewer w3ua, we appreciate your encouragement and positive feedback. We are pleased that your concerns have been addressed. We will improve our work according to your suggestions. In addition, our dataset, source code, and project website will be released publicly. Thank you once again for your valuable time and effort in enhancing our paper.

---

### Author Rebuttal · Authors · 2024-08-06

Dear Reviewers and Area Chairs,

We would like to thank you for your valuable time and efforts in providing these insightful questions and suggestions for improving our paper. We are also pleased that the reviewers have generously appreciated our new MVUAV dataset and the semi-supervised MVSS baseline - SemiMV.

In the individual response, we have provided detailed, point-by-point responses to each reviewer's comments. A one-page PDF file is also attached that contains all relevant figures and tables used in the response, including a figure illustrating the feature transformation process and corresponding dimension changes of Eqs. 4 & 5 & 6 (Reviewer XBRU), two ablation tables verifying the impact of hyperparameters (Reviewer XBRU), and a table comparing our MVUAV dataset with related UAV datasets (Reviewer Mx7S). We hope this could provide a more comprehensive presentation of our work.

If there are any further questions, please let us know. We would be happy to discuss and answer them in the reviewer-author discussion phase.

Best Regards,
Authors of Paper 906

---

### Decision · Program_Chairs · 2024-09-25

**Decision:**

Accept (poster)

**Comment:**

This paper proposes a new benchmark dataset, MVUAV, and a semi-supervised learning approach for multispectral video semantic segmentation (MVSS) for UAV images. The initial ratings are mixed. The authors provided a strong rebuttal, and all reviewers increased their rating after the discussion, reaching an all-positive consensus. The authors should improve the representation and explanations, clarify the motivation and significance of research on UAV-view images, address the ethical concerns, and incorporate the additional feedback provided during the rebuttal and discussion period.